# Coherent Integration of Organic Gel Polymer Electrolyte and Ambipolar Polyoxometalate Hybrid Nanocomposite Electrode in a Compact High-Performance Supercapacitor

**DOI:** 10.3390/nano12030514

**Published:** 2022-02-01

**Authors:** Jun-Jie Zhu, Luis Martinez-Soria, Pedro Gomez-Romero

**Affiliations:** 1Catalan Institute of Nanoscience and Nanotechnology (ICN2), CSIC and BIST, Campus UAB, Bellaterra, 08193 Barcelona, Spain; luis.martinezsoria@icn2.cat; 2Consejo Superior de Investigaciones Científicas (CSIC), 28006 Madrid, Spain

**Keywords:** gel polymer electrolytes, polyoxometalates, supercapacitors, hybrid electrodes, organic electrolytes, impedance spectroscopy

## Abstract

We report a gel polymer electrolyte (GPE) supercapacitor concept with improved pathways for ion transport, thanks to a facile creation of a coherent continuous distribution of the electrolyte throughout the electrode. Poly(vinylidene fluoride-co-hexafluoropropylene) (PVDF-HFP) was chosen as the polymer framework for organic electrolytes. A permeating distribution of the GPE into the electrodes, acting both as integrated electrolyte and binder, as well as thin separator, promotes ion diffusion and increases the active electrode–electrolyte interface, which leads to improvements both in capacitance and rate capability. An activation process induced during the first charge–discharge cycles was detected, after which, the charge transfer resistance and Warburg impedance decrease. We found that a GPE thickness of 12 μm led to optimal capacitance and rate capability. A novel hybrid nanocomposite material, formed by the tetraethylammonium salt of the 1 nm-sized phosphomolybdate cluster and activated carbon (AC/TEAPMo12), was shown to improve its capacitive performance with this gel electrolyte arrangement. Due to the homogeneous dispersion of PMo12 clusters, its energy storage process is non-diffusion-controlled. In the symmetric capacitors, the hybrid nanocomposite material can perform redox reactions in both the positive and the negative electrodes in an ambipolar mode. The volumetric capacitance of a symmetric supercapacitor made with the hybrid electrodes increased by 40% compared to a cell with parent AC electrodes. Due to the synergy between permeating GPE and the hybrid electrodes, the GPE hybrid symmetric capacitor delivers three times more energy density at higher power densities and equivalent cycle stability compared with conventional AC symmetric capacitors.

## 1. Introduction

Portable/wearable electronic devices (e.g., mobile phones, laptops, cameras, smartwatch, activity trackers, and many more) are flourishing and demanding improved energy storage devices: small, thin, lightweight, flexible, fire-retardant, etc. [1]. Gel polymer electrolytes can simultaneously address many of these demanding challenges. They are fire-retardant, their geometry shape is variable, and they allow for fabricating safe, thin, and flexible devices.

Gel polymer electrolytes (GPE) are composed of polymer matrices and supporting electrolytes. Many polymers could serve as the matrices, such as poly(vinyl alcohol) (PVA), poly(methyl methacrylate) (PMMA), poly(ethylene oxide) (PEO), and copolymer poly(vinylidene fluoride-hexafluoro propylene) (PVDF-HFP)) [2]. Depending on their hydrophilic or hydrophobic nature, these polymers are suitable for aqueous or nonaqueous electrolytes. Compared to the conventional cell configuration consisting of liquid electrolytes and separators (e.g., cellulose membrane, Celgard membrane), polymer matrices in GPEs play several roles: as an effective separator, a reservoir for electrolytes, and pathways for ion diffusion. Due to the integrated functions and merits, GPE has widely substituted liquid electrolytes in a variety of energy conversion and storage devices, including solar cells [3], lithium-ion batteries [4,5], supercapacitors [6,7], etc.

Supercapacitors possess a high power density and long cycle life, but their energy density has yet to be improved compared with batteries. Electrodes, more specifically, the active materials on electrodes, are the critical components of supercapacitors, which could determine the intrinsic capacitance. Carbon-based materials can store energy in the electric double layer and have become the widest studied and commercialized active material for supercapacitors. The capacitance of carbon-based materials can gain more capacitance by introducing the other energy storage mechanism: redox reaction. Decorating with heteroatoms [8,9], incorporating redox-active materials could take the capacitance of carbon-based hybrid materials to the next level.

Polyoxometalates (POMs) are transition metal oxide clusters of around 1 nm in size, capable of delivering fast reversible multi-electron redox reactions, making them a competitive candidate in various energy storage devices [10]. For supercapacitors, POMs usually do not act as the active materials alone since the ion diffusion rate in crystallized POMs is sluggish, contrary to the high-power characteristic of supercapacitors. On the other hand, POMs can carry out their fast reversible redox reaction when they are well-dispersed. Thus, spreading or anchoring POMs on other materials, including activated carbon [11,12], reduced graphene oxide [13,14,15], carbon nanotubes [16,17], and a metal–organic framework [18] would be a solution. POMs-based hybrid materials have already shown their potential in supercapacitors to enhance capacitance and energy densities.

Since energy density is proportional to the square of the operating voltage window, using the nonaqueous electrolytes of the larger stable potential window is always favorable. The copolymer PVDF-HFP is compatible with many nonaqueous electrolytes, and so could serve as the matrix for nonaqueous gel polymer electrolytes. Conventional organic electrolytes [19], redox-active electrolytes [20], and ionic liquids [21,22] have already been incorporated with PVDF-HFP matrix to fabricate flexible supercapacitors with high energy density and safety.

Compared with liquid electrolytes, gel polymer electrolytes composed of the same liquid electrolytes typically show lower ionic conductivity because the polymer matrix impedes free ion movement [23,24], which affects the rate capability of supercapacitors. The rate performance of GPE supercapacitors can hardly exceed that of liquid-electrolyte supercapacitors.

Nonetheless, we can tackle this problem from an engineering perspective. Besides the nature of active materials and electrolytes, designing specific pathways for ion diffusion to reduce the diffusion route can improve the electrochemical performance of energy storage devices.

Herein, we introduce a well-engineered coherent electrode–electrolyte interface by integrating the copolymer PVDF-HFP throughout electrodes and the electrolyte. We obtain the thin gel polymer electrolyte once we soak the PVDF-HFP matrix in 1 M TEABF_4_/acetonitrile. The coherent PVDF-HFP matrix provides a highway for ion movement throughout the electrodes and the electrolyte. This structure enhances the rate capability and the capacitance, which, in turn, allows for both high power densities and energy densities. Furthermore, we fabricate a symmetric supercapacitor with high energy density and power density thanks to the novel activated carbon/polyoxometalates hybrid material coupled with the gel polymer electrolyte.

## 2. Experimental Section

### 2.1. Materials

Activated carbon was purchased from Cabot Corporation, Alpharetta, GA, USA. Aluminium foil (>99%, 18 μm) was purchased from Goodfellow, Hamburg, Germany. Acetonitrile (electronic grade), N-methyl-2-pyrrolidone (anhydrous, ≥99.5%), dimethylformamide (anhydrous, ≥99.8%), tetraethylammonium tetrafluoroborate (≥99%), Poly(vinylidene fluoride-co-hexafluoropropylene) (average molecule weight ≈ 400,000), poly(vinylidene fluoride) (average molecule weight ≈ 534,000), phosphomolybdic acid (HPMo12), and tetraethylammonium chloride (≥98%) were from Sigma-Aldrich without additional purification. Nitrocellulose filter membranes (pore size = 0.025 mm, thickness = 0.17 mm) were purchased from MF-Milipore, Merck, Madrid Spain. Celgard 2400 membranes were purchase from Celgard, LLC, Charlotte, NC, USA.

### 2.2. Preparation of Gel Electrolytes

First, 2 g of PVDF-HFP was added in 10 mL acetone with vigorous stirring at 50 °C until a transparent viscous solution formed. Subsequently, the solution was cast on a glass plate using the doctor-blade method. After a few minutes, the acetone evaporated, and a white, half-transparent film was obtained. The film can be easily peeled off by a spatula and stored in an argon-filled glove box for further use. We can obtain PVDF-HFP films of different thicknesses by selecting separate doctor-blade bars, ranging from 2 μm to 36 μm. The film would be soaked in the liquid electrolyte just before assembling the devices.

### 2.3. Synthesis of Hybrid Materials

The organic–inorganic POM salt (TEAPMo12) was synthesized through metathesis reaction, as our previous literature reported. Typically, 100 mL of a 20 mM aqueous solution of phosphomolybdic acid was added to 100 mL of a 70 mM aqueous solution of tetraethylammonium chloride. The slight yellow precipitate appeared immediately. The suspension was kept stirring for 6 h, filtered, washed with DI water, collected, and dried at 120 °C overnight. Tetraethylammonium phosphomolybdate (TEAPMo12, [(C_2_H_5_)_4_N)]_3_PMo_12_O_40_) was obtained.

The hybrid material AC/TEAPMo12 were prepared by sonicating 0.5 g AC in 100 mL of 10 mM TEAPMo12 in DMF for 6 h. The samples were collected by filtration and dried at 120 °C overnight.

### 2.4. Preparation of Electrodes

The conventional electrodes were fabricated by mixing the active material (AC), carbon black and poly(vinylidene fluoride) at a weight ratio of 85:5:10. The mixture was first formed as a slurry by adding a few drops of N-methyl-2-pyrrolidinone (NMP) and then coated onto 18 μm aluminum foil and dried under vacuum at 120 °C for 12 h.

The gel-polymer-contained electrodes were prepared by dispersing the active materials, carbon black and PVDF-HFP (pre-dissolved in NMP), at a weight ratio of 85:5:10. The slurry was stirred for 6 h and then coated onto aluminum foil and dried under vacuum at 120 °C for 12 h. The thickness of the coated AC layer on Al foil was around 47 μm, and the thickness of coated AC/TEAPW12 was around 36 μm. The loading masses of both electrodes were around 2.4 mg·cm^−2^.

### 2.5. Characterization

Thermogravimetric analyses were carried out with NETZSCH-STA 449 F1 Jupiter thermal analysis system under oxygen flow with a heating rate of 10 °C min^−1^ from room temperature to 900 °C. Scanning electron microscopy images were taken on Quante 650 FEG microscopy. High angle annular dark field scanning transmission electron microscopy (HAADF-STEM) images were taken on FEI Tecnai G2 F20 microscopy.

Three types of cells were fabricated for different characterization. T-type stainless steel Swagelok^®^ cells were used to characterize the materials in the three-electrode configuration. Ag/Ag^+^ (0.01 M AgNO_3_) was used as the reference electrode. An electrode loaded with the treble weight of the AC active material was used as the counter electrode. CR2032 coin cells were used to fabricate symmetric supercapacitors. All the electrodes were pressed at 3 MPa before assembling. Additionally,1 M tetraethylammonium tetrafluoroborate (TEABF_4_) in acetonitrile severed as the electrolyte. All the cells were assembled in an argon-filled glovebox (Jacomex GP with O_2_ < 5 ppm and H_2_O < 5 ppm). Cyclic voltammetry (CV) was performed in a three-electrode configuration at various scan rates to investigate the capacitive behavior of individual electrodes. Cyclic polarization (CP), electrochemical impedance spectra (EIS), and galvanostatic charge–discharge (GCD) were performed in a two-electrode cell to evaluate the capacitive performance in devices and cycling stability. All the electrochemical tests were conducted on Biologic VMP3 multi-channel potentiostat. The calculation of capacitance, power densities and energy densities are presented in Appendix A.

## 3. Results and Discussion

### 3.1. Evaluation and Optimization of GPE

We prepared PVDF-HFP films of various thicknesses, ranging from 2 to 36 μm. Figure 1a presents photographs of the dry PVDF-HFP films. The thinnest film (2 μm) is transparent, and the rest are translucent. The Fourier-transform infrared spectra of PVDF-HFP films are presented in Appendix A. All the films turned to transparent gel after being soaked in 1 M TEABF_4_ acetonitrile electrolyte (Figure 1b). After that, we obtained gel polymer electrolytes.

To construct a coherent electrode–electrolyte interface, PVDF-HFP serves as the binder in the electrodes. Figure 1c presents the morphology of PVDF-HFP film. We can discern micro-scale PVDF-HFP crystals connecting. The pores among PVDF-HFP crystals serve as the reservoir for the electrolyte and pathway for ion migration. Figure 1d,e present the AC electrode with PVDF and PVDF-HFP, respectively. The two kinds of electrodes show a similar morphology: AC particles dispersed and connected by Carbon SuperP (Csp) tiny particles. We did not find any agglomeration of PVDF-HFP in Figure 1e. PVDF-HFP is soluble in NMP, which allows it to be dispersed homogeneously in the whole slurry before coating the electrodes. After drying the electrodes, PVDF-HFP covers the AC and Carbon SuperP particles, serving as an effective binder. Figure 1f presents the cross-section SEM image of the electrode–electrolyte interface. Many AC and Csp particles stick on the PVDF-HFP film after cutting, indicating the excellent compatibility between the electrodes and the films.

To evaluate the influence of GPE on capacitive performance, we built AC symmetric supercapacitors and performed electrochemical characterization. The electrochemical impedance analysis could give us some insight into the electrode/electrolyte interface, capacitive behaviors, ion diffusion, etc. Figure 2a,b present the Nyquist plots of the fresh and activated AC symmetric supercapacitors. The 2 μm PVDF-HFP film device is not included because the 2 μm PVDF-HFP film could not separate the two electrodes effectively. The thickness threshold for this kind of effective gel polymer electrolyte and separator would be around 5 μm.

All the spectra are fitted by an equivalent circuit shown in Figure 2c. At the very high-frequency region, the intercepts at real part Z, represented by R_s_, provide the combined effect of the ionic resistance of the electrolyte, the intrinsic resistance of the active materials, and the contact resistance at the active material/current collector interface. Since all the cells have the same active material, the current collector and the liquid electrolyte, the intercepts are roughly the same at very high frequency. The semicircles at high frequency represent a parallel combination of charge-transfer resistance R_ct_ and the double layer capacitance C_dl_. The following transitory parts between the semicircles and the final linear parts represent the Warburg impedance. This relates to the diffusive resistance of the electrolyte in the electrodes. Considering the different slopes of the final linear parts, we apply a modified restricted diffusion element W_s_ for fitting [25]. It has three parameters (Equations S6 and S7 in Appendix A). W_s_-R represents the diffusion impedance; W_s_-T represents the diffusion time; W_s_-P is an exponential factor: the number 0.5 means the infinite Warburg diffusion (slope = 1), and 1.0 means finite Warburg diffusion (vertical line) [26,27].

Table 1 lists the fitting results of the impedance spectra. The diameter of the compressed semicircle (R_ct_) shows a significant difference among the cells. For the fresh cells with GPE, R_ct_ increases with the increase in the thickness of PVDF-HFP films. The R_ct_ values are in this order: 5 μm < cellulose (thickness = 170 μm and pore size = 25 μm) < 12 μm < 36 μm < celgard (thickness = 25 μm). R_ct_ normally involves the charge transfer process stem from redox-active materials. However, in AC supercapacitors, many studies found the R_ct_ exists too [24,28,29]. Since the pseudocapacitance in the activated carbon is insignificant, the R_ct_ is mainly dominated by the adsorption/desorption of ions, or in other words, pore accessibility [29]. The R_ct_ of the cells with the liquid electrolyte and the conventional separators (Celgard membrane and cellulose paper) do not change significantly after cycling. By contrast, R_ct_ of GPE cells decreased remarkably after cycling. The R_ct_ of the cell with 12 μm GPE decreases from 31 Ω to 2.8 Ω. This phenomenon implies an activation process at the electrode–electrolyte interface. The electrolyte has easier access to the pores after the activation. The PVDF-HFP serves as both the binder and the matrix for the gel electrolyte. At first, the TEABF_4_ in acetonitrile only wet the PVDF-HFP film, not the PVDF-HFP in the electrodes. With the charge–discharge process ongoing, the PVDF-HFP in the electrodes infuses with TEABF_4_, facilitating adsorption/desorption of ions in pores, and the porous structure remains.

Moreover, the Warburg impedance (both W_s_-R and W_s_-T) of GPE cells decreases after cycling, too, further confirming an improved ion-diffusion process. By contrast, W_s_-R and W_s_-T of Celgard membrane and cellulose paper cells do not show as significant a change as GPE cells do after cycling, revealing that the decrease in W_s_-R and W_s_-T is mainly associated with PVDF-HFP gel. The activation process happens when PVDF-HFP either exists in the electrodes or gel electrolytes (Appendix A). W_s_-P is an indicator to distinguish whether the diffusion is finite. The W_s_-P of the Celgard cell is close to 0.5, corresponding to an infinite diffusion that mainly happens in the bulk electrolyte. For 12 μm and 36 μm GPE-based cells, we discern a clear increase (from 0.7 to 0.8) in W_s_-P after cycling, revealing that the diffusion behavior changes during cycling. It agrees well with the deduction derived from R_ct_. With charge–discharge going, more electrolytes permeate the pores in the PVDF-HFP, and the diffusion behavior shifts from infinite diffusion in bulk electrolyte to finite diffusion in the porous structure. Additionally, 5 μm GPE cells exhibit a slighter change in W_s_-P after activation, which must relate to their thinnest thickness. The porous structure of 5 μm GPE cells is permeated quickly; thus, the diffusion behavior is mainly finite initially. The variations of these parameters (the decrease in R_ct_, W_s_-R and W_s_-T; the increase in W_s_-P) increase with the thickness of GPEs, revealing that the thicker GPE cells undergo a more remarkable activation.

The impedance analysis confirms the existence of the activation process in GPE cells. The diffusion is promoted after the activation, and more pores become accessible to the electrolyte. Concerning R_ct_ and W_s_, the GPE cells are superior to the conventional configuration cells. Next, we compare their capacitive performance.

Figure 2d shows the cyclic polarization curves of the cells at 100 mV·s^−1^. The CP curves of the cells with cellulose or Celgard separator in the liquid electrolyte are not perfectly rectangular in shape. Their parallelogram shape implies their resistive nature. By contrast, the cells of 5 μm and 12 μm present the CP curves of the ideal rectangle shape, indicating their pure capacitive behavior even at the high scan rate of 100 mV·s^−1^. The difference among the CP curves agrees with the previous impedance analyses: GPE cells have smaller R_ct_ and Warburg impedance; thus, they remain capacitive at high scan rates.

Figure 3e presents galvanostatic charge–discharge curves at 8 A·g^−1^ of the cells. All the curves give a triangle shape, indicating the cells’ capacitive nature. The GPE cells have much smaller IR drops, which is another indicator of low resistance in GPE cells. Figure 2f lists the specific capacitance of the cells at various scan rates. The specific capacitance of the GPE cells is almost equivalent to or slightly higher than that in the conventional configuration cell (a cellulose separator and the liquid electrolyte) at low scan rate (2 mV· s^−1^). It reveals that the PVDF-HFP in electrodes and separators has no detriments on the capacitive performance. Instead, the capacitance of the 12 μm GPE cell is improved. We speculate that the PVDF-HFP on the active material allows the electrolyte to pass through, and some PVDF-HFP-covered areas are still active, while PVDF-covered areas are mostly inert. Even though the cells show only a slight difference in specific capacitance at the low scan rate (2 mV·s^−1^), the difference between the GPE cells and the cells with conventional separators and liquid electrolyte becomes remarkable the increase in scan rates. The excellent rate capability of the GPE cells must be ascribed to the coherent integration of PVDF-HFP at the electrode–electrolyte interface.

Considering the capacitance and rate capability, we select the 12 μm GPE as the best option. The as-prepared PVDF-HFP film is much thinner than PVDF-HFP GPE in other studies [4,5,6,22], which implies its potential for compact devices. Our previous study has proved that the organic–inorganic POM salt can enhance the volumetric capacitance of activated carbon in the organic electrolyte, suitable for compact devices. Thus, in the following part, we develop a novel AC/POMs hybrid material (AC/TEAPMo12) to enhance the volumetric capacitive performance of the device further.

### 3.2. Evaluation of the Hybrid Material

XRD patterns confirm the AC/TEAPMo12 contains TEAPMo12 (Appendix A). The composition of the hybrid materials AC/TEAPMo12 was determined by thermogravimetric analyses (Figure 3a). The pristine AC suffers a dramatic weight loss since 600 °C, which refers to the combustion of carbon in the air. Finally, 2.12 wt% of its initial weight remains as ashes. Pure TEAPMo12 undergoes a significant weight loss between 300 °C to 400 °C due to the decomposition of tetraethylammonium moieties. Subsequently, its weight stays relatively stable (81.1 wt% of the initial weight) until 730 °C. The hybrid AC/TEAPMo12 shows a weight loss before 620 °C, followed by a plateau that covers the range from 630 °C to 730 °C. At this plateau, 28.5 wt% of the initial weight remains. Therefore, we can estimate that AC/TEAPMo12 contains 33.3 wt% of TEAPMo12. TEAPMo12 possesses a large proportion in AC/TEAPMo12, but the XRD pattern of AC/TEAPMo12 only presents weak peaks from TEAPMo12 because TEAPMo12 clusters are not well crystallized, but nanocrystals are.

Figure 3b presents the SEM images of AC/TEAPMo12, from which we cannot see any aggregation of TEAPMo12. (The transmission electron microscopy images of AC and AC/TEAPMo12 are presented in Appendix A) We expect the TEAPMo12 clusters to spread homogeneously on AC substrate as other keggin-type POMs do [11,12,30]. To study the distribution characteristics of TEAPMo12, we observe the samples under scanning transmission electron microscopy in the high angle annular dark field (HAADF-STEM). Figure 3c shows the HAADF-STEM image of pristine AC. It reveals the porous nature of AC. Plenty of micropores (pore size < 2 nm) and mesopores (pore size from 2 to 50 nm), corresponding to the dark areas in the image, can be discerned. By contrast, in the HAADF-STEM image of AC/TEAPMo12 (Figure 3d), we cannot distinguish the pores. Instead, numerous bright dots around 1 nm spread on the sample, corresponding to Mo in TEAPMo12 clusters. It confirms that the TEAPMo12 clusters are dispersed homogeneously in nanoscale on AC substrate, agreeing with the speculation from XRD patterns. Indeed, the anchoring of TEAPMo12 clusters on AC would lead to a slight decrease in pore volume and surface area of the hybrid materials [11], and the bright dots make the contrast between pores and matrix less visible.

This hybrid material is characterized in three-electrode configurations to study its electrochemical behaviors and two-electrode symmetric cells to explore its performance in practice. Figure 4a compares the CVs of AC and AC/TEAPMo12 at 5 mV·s^−1^ in the full potential range. The CV of AC/TEAPMo12 exhibits four pairs of redox waves in addition to the rectangle shape of AC, confirming the electrochemical activities of TEAPMo12 clusters.

The charge–storage mechanism of AC/TEAPMo12 was analyzed by discriminating the non-diffusion-controlled contribution and diffusion-controlled contribution to the total charges stored in AC/TEAPMo12 electrodes. The dependence of the current response on the scan rate reveals the charge–storage mechanism, as shown in Equation (1)
(1)$$i={\mathrm{av}}^{b}$$
where i is current obtained at a specific scan rate v, and a and b are adjustable parameters. The current response of linear dependence on scan rate (b = 1) usually means the charge is stored by a fast response mechanism, such as surface capacitive mechanism and redox reactions that could perform fast rate [31]. Otherwise, if any slow diffusion of ion or electron transfer limits the reaction, the dependence could deviate from linearity (b < 1). We performed CVs at various scan rates (Appendix A) and ran the linear fitting at the potentials where the redox reactions happen according to Equation (2)
(2)log i = log a + b log v

Figure 4b,c show the fitting curves. Table 2 lists the b values at the redox potentials. All the R^2^ are close to 1, revealing the effective fitting. The b values at all the redox potentials are close to 1, which means the non-diffusion-controlled process dominates the charge storage. It might seem surprising for redox-active materials, but this phenomenon happens when the redox-active species are well-dispersed, such as POMs on 3D framework [31], AC/TEAPW12 hybrids [11]. The contribution of the non-diffusion-controlled process and the diffusion-controlled process to the capacitance in the whole potential is analyzed and presented in Appendix A. We conclude that the non-diffusion-controlled process dominates the energy storage in the whole potential range at any scan rate.

### 3.3. Evaluation of Symmetric Devices

We put two identical hybrid electrodes and a Ag/Ag^+^ reference electrode in a T-type Swagelok cell to explore its behaviors in symmetric capacitors. After activation, the electrodes in the symmetric capacitor deliver a potential of zero-charge (when the voltage of the cell is zero) around −0.65 vs. Ag/Ag^+^ during galvanostatic charging–discharging from 0–2.7 V (Figure 4d). The variation of the potential of the positive electrode is from −0.65 V to 0.75 V. That of the negative electrode is from −0.65 V to −1.9 V. This potential of zero-charge is in the middle of the four pairs of redox waves, which means the positive and the negative electrodes can carry out two redox reactions, respectively. The cyclic voltammograms of the two electrodes in their working potential range exhibit two redox pairs, respectively (Figure 4e). It confirms that all the redox reactions in Figure 4a participate in energy storage in the symmetric capacitor. Figure 4f compares the cyclic polarization curves of AC and AC/TEAPMo12 in two-electrode symmetric capacitors. The integrated area of the AC/TEAPMo12 cell curve is larger than that of AC cell, indicating the volumetric capacitance (*Y*-axis in Figure 4f is normalized by volume). The CP curve exhibits two redox waves because some redox waves on the positive and negative electrodes overlap and merge into one wave. It is further evidence that redox reactions are involved in the energy storage process of the symmetric capacitor. The CP curves of AC-Cellulose, AC-GPE and AC/TEAPMo12-GPE symmetric capacitors at various scan rates are presented in Appendix A 5, Appendix A, and the capacitance calculated from the CP curves are presented in Appendix A.

We perform galvanostatic charging–discharging at various current densities to explore the capacitive performance of the conventional AC symmetric capacitor (AC-Cellulose), the GPE AC symmetric capacitor (AC-GPE), and the GPE AC/TEAPMo12 symmetric capacitor (AC/TEAPMo12-GPE) for practice use. The charging–discharging curves are presented in Appendix A. Figure 5a presents the gravimetric capacitance of the symmetric capacitors. The gravimetric capacitance of AC-GPE and AC/TEAPMo12-GPE are almost equivalent at various current densities. It is not surprising that the hybrid material AC/TEAPMo12 does not enhance the gravimetric capacitance because the TEAPMo12 molecule has a large weight (>2000 g·mol^−1^). Since POMs are compact nanoclusters, they are more effective in improving volumetric capacitance. At high current densities, the enhancement of capacitance in GPE cells is significant and must be assigned to the coherent integration between the electrode–electrolyte interfaces as we mentioned above. The volumetric capacitance of AC/TEAPMo12 is remarkably improved (Figure 5b). At 0.5 A·g^−1^, AC/TEAPMo12 delivers a volumetric capacitance of 74.6 F ·cm^−3^, 41% larger than AC with the same gel polymer electrolyte. At any current densities, the volumetric capacitance of AC/TEAPMo12 gains more than 40 % improvement. The POMs clusters are mainly absorbed in pores and contribute little to the whole volume but a lot to the total weight [11]. Thus, the gravimetric capacitance improves only slightly, but the volumetric capacitance improves significantly. The hybrid material is suitable for compact devices.

The ragone plot (Figure 5c) compares the energy densities and power densities of the three symmetric capacitors. To better view the practical application, we normalize the densities by the total volume of the electrodes (including current collectors), the separators, and the electrolytes. Since the thin PVDF-HFP film serves as both the electrolyte and the separator after soaking, the AC-GPE capacitor deliver two times more densities energy (2.86 mWh·cm^−3^ at 1.77 W·cm^−3^ compared with 0.87 mWh·cm^−3^ at 0.68 W·cm^−3^) at even high power densities. The energy densities and power densities are further improved once AC/TEAPMo12 serves as the active material in the electrodes. It could deliver energy densities as high as 3.88 mWh·cm^−3^ at 2.39 W·cm^−3^. Thanks to the gel polymer electrolyte and the hybrid active material, compared with the conventional configuration, the AC/TEAPMo12-GPE symmetric capacitor exhibits superior energy densities and power densities.

Figure 5d shows the cycle stability of the three capacitors. All the cells suffer a slight drop in capacitance for the first few cycles, followed by a steady and prolonged loss in capacitance. After 10,000 cycles, all the cells have more than 90% of their initial capacitance. It proves that the PVDF-HFP gel polymer electrolyte and the AC/TEAPMo12 active material do not have detrimental effects on their cycle stability. Finally, coulombic efficiency of our AC/TEAPMo12-GPE cell in the first cycle was 97%, went up to 99.0% in the third cycle, and eventually stayed steady at 99.5% until the end of the 10,000 cycles.

## 4. Conclusions

In this work, we developed a facile way to build coherent pathway throughout the electrodes and the gel polymer electrolyte (separator) by applying PVDF-HFP both in electrodes and electrolyte. The AC symmetric capacitor with GPE show that 5 μm GPE is the threshold as effective separator, and 12 μm GPE is the optimized thickness with regard to the capacitance and rate capability. Compared with the AC symmetric capacitors of conventional configuration, the GPE-based AC symmetric capacitors exhibit slightly larger capacitance and much better rate capability, which must be assigned to the coherent integration throughout the electrode and the electrolyte.

To further improve the capacitive performance of this GPE-based symmetric capacitor, we hybridize AC with TEAPMo12 and apply this hybrid material in a symmetric capacitor. TGA confirms that the AC/TEAPMo12 contains 33.3 wt% of TEAPMo12. HAADF-STEM show that TEAPMo12 clusters are dispersed homogeneously at nanoscale. Owing to the good dispersion, the redox reactions of TEAPMo12 are non-diffusion controlled, favoring high rate capability. The zero-charge potential of the AC/TEAPMo12 is in the middle of the four pairs of redox wave, which allows the positive and negative electrode in symmetric capacitor to perform two redox waves, respectively. The AC/TEAPMo12 symmetric capacitor could deliver a much higher energy (around 4.5 times) at a much higher power density (around 3.5 times) compared with the AC symmetric capacitor of the conventional configuration without sacrificing any cycle stability.

## Figures and Tables

**Figure 1 nanomaterials-12-00514-f001:**
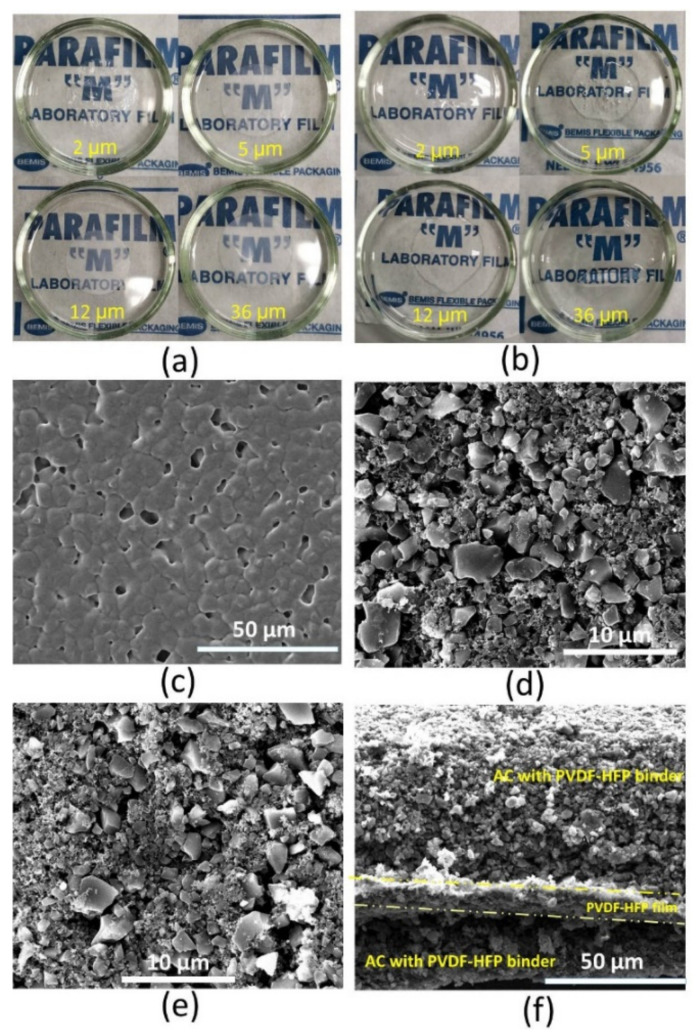
Photographs of PVDF-HFP films of various thicknesses: (**a**) dry films and (**b**) films soaked with 1 M TEABF_4_ in acetonitrile. SEM images: (**c**) 12 μm PVDF-HFP film, (**d**) activated carbon electrodes with PVDF binder, (**e**) activated carbon electrodes with PVDF-HFP binder and (**f**) cross-section of the electrode/PVDF-HFP film/electrode sandwich (PVDF-HFP as the binder in the electrodes).

**Figure 2 nanomaterials-12-00514-f002:**
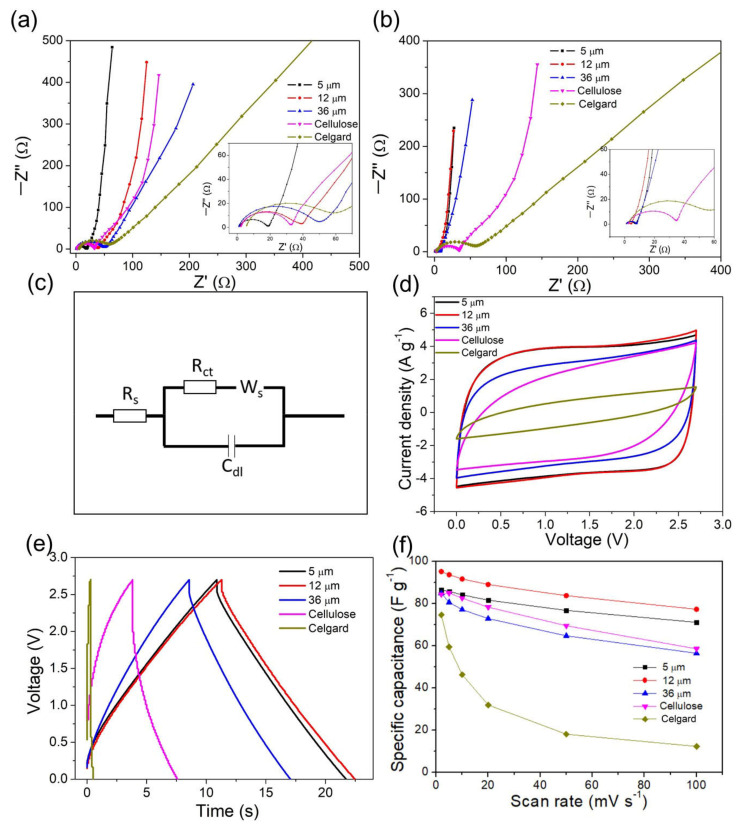
Nyquist plot of the cells with 5 μm, 12 μm, and 36 μm PVDF-HFP gel polymer electrolyte, or cellulose and Celgard separator in the liquid electrolyte: (**a**) the fresh cell, (**b**) the activated cells. (the intercepts and semicircles in the high-frequency range are zoomed in the inset graphs). (**c**) The equivalent circuit for fitting the impedance spectra. (**d**) Cyclic polarization curves at 100 mV·s^−1^ and (**e**) Galvanostatic charge–discharge curves at 8 A·g^−1^ of the cells with 5 μm, 12 μm, and 36 μm, PVDF-HFP gel polymer electrolyte, or cellulose and Celgard separator in the liquid electrolyte. (**f**) The specific capacitance of the cell at various scan rates.

**Figure 3 nanomaterials-12-00514-f003:**
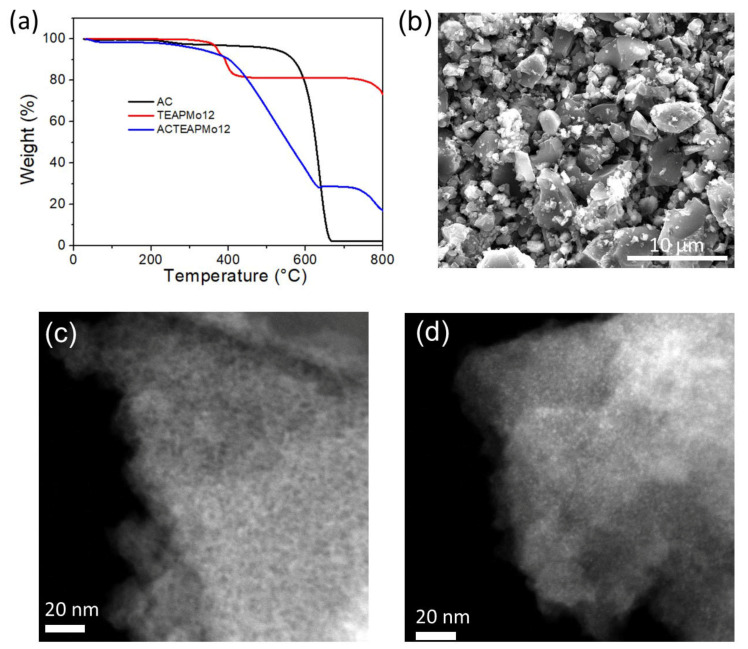
(**a**) The thermogravimetric curves of AC, TEAPMo12 and AC/TEAPMo12. (**b**) SEM image of AC/TEAPMo12. HAADF-STEM images of (**c**) AC and (**d**) AC/TEAPMo12.

**Figure 4 nanomaterials-12-00514-f004:**
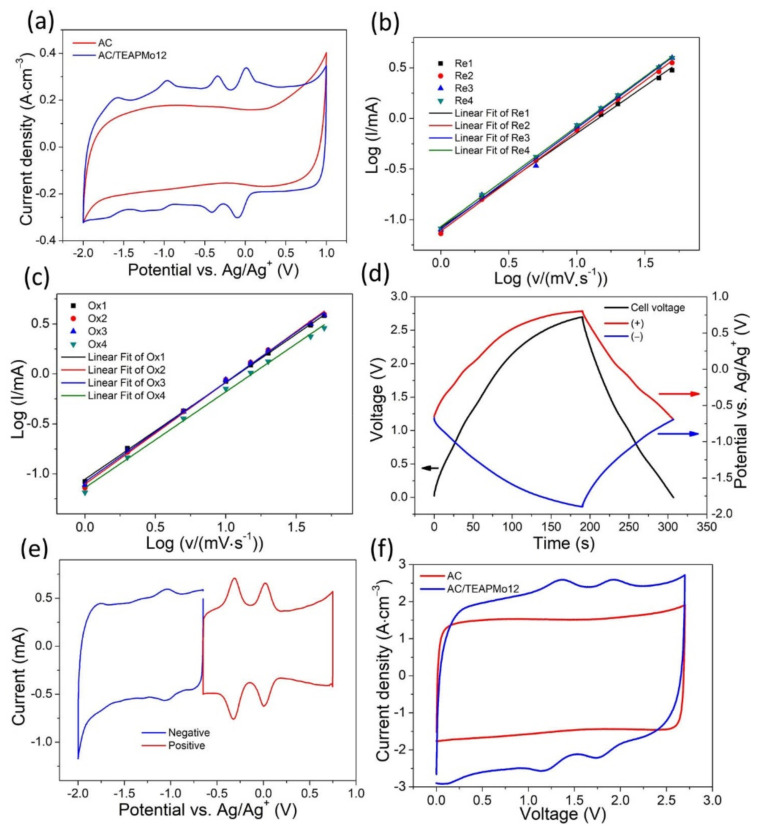
(**a**) CVs of AC and AC/TEAPMo12 at 20 mV·s^−1^ in three-electrode configuration. b-value determination for (**b**) oxidation peaks and (**c**) reduction peaks of CVs of AC/TEAPMo12. (**d**) The variation of the potential of the positive electrode, the negative electrode, and the voltage of the symmetric capacitor during galvanostatic charging–discharging. (**e**) CVs of the positive and the negative electrodes in their potential range. (**f**) Cyclic polarization curves of AC and AC/TEAPMo12 at 100 mV·s^−1^.

**Figure 5 nanomaterials-12-00514-f005:**
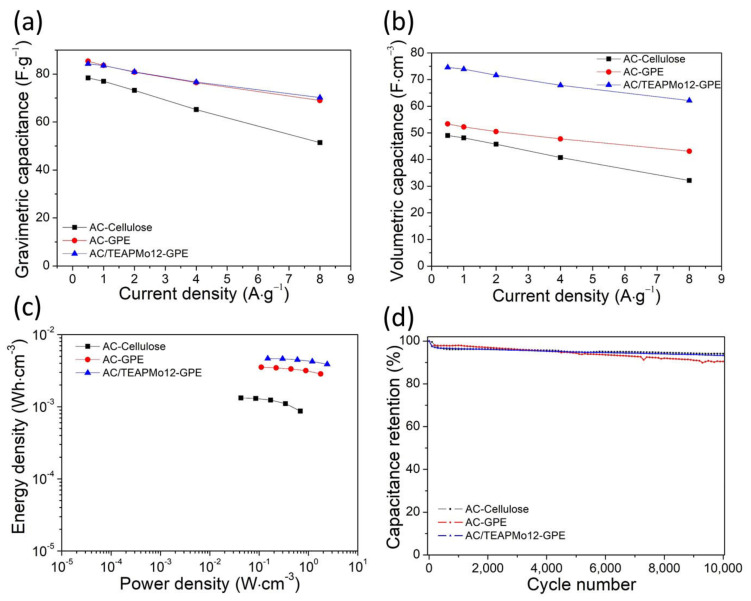
(**a**) Gravimetric capacitance, (**b**) volumetric capacitance, (**c**) ragone plot, and (**d**) cycle stability of AC-cellulose (with the cellulose separator and the liquid electrolyte), AC-GPE (AC with the 12-μm gel polymer electrolyte), and AC/TEAPMo12-GPE (AC/TEAPMo12 with the 12 μm gel polymer electrolyte) symmetric cells.

**Table 1 nanomaterials-12-00514-t001:** Some fitting results of impedance spectra.

Cells	State	R_s_/Ω	C_dl_/mF	R_ct_/Ω	W_s_-R/Ω	W_s_-T/s	W_s_-P
5 μm	fresh	1.12	1.83	15.78	10.03	0.21	0.85
	activated	1.08	3.76	6.51	1.46	0.04	0.87
12 μm	fresh	1.87	1.93	32.16	22.01	0.32	0.74
	activated	0.83	4.17	3.23	2.10	0.06	0.83
36 μm	fresh	0.89	2.32	44.24	29.7	0.42	0.74
	activated	1.08	3.54	6.53	1.63	0.03	0.84
Cellulose	fresh	2.13	2.81	29.44	7.60	0.04	0.69
	activated	2.14	4.36	28.83	9.48	0.06	0.66
Celgard	fresh	2.09	1.55	31.23	91.56	0.13	0.58
	activated	2.23	1.19	42.62	45.47	0.05	0.54

**Table 2 nanomaterials-12-00514-t002:** Values of b Equations (1) and (2) and regression coefficients (R^2^) derived from the fitting current of reduction peaks (Re) and oxidation peaks (Ox) at various scan rates.

	Re1	Ox1	Re2	Ox2	Re3	Ox3	Re4	Ox4
b	0.932	0.967	0.990	1.000	0.999	0.998	0.988	0.960
R^2^	0.997	0.999	0.997	0.999	0.997	0.999	0.998	0.999

## Data Availability

The data presented in this study are available on request from the corresponding authors.

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
