# Peer review of "Coherent Integration of Organic Gel Polymer Electrolyte and Ambipolar Polyoxometalate Hybrid Nanocomposite Electrode in a Compact High-Performance Supercapacitor"

_nanomaterials, 2022, doi:10.3390/nano12030514_

Round 1
Reviewer 1 Report
In this article, nanocomposite electrode composed of activated carbon and TEAPMo12 was successfully fabricated and assembled into a symmetric supercapacitor. Meanwhile, author innovatively regarded gel polymer (PVDF-HFP) as binder which was used in the process of preparing electrodes and electrolyte matrix. Compared to conventional binder (PVDF) and membranes (Celgard and Nitrocellulose membranes), this method facilitated ion transport throughout the electrodes and the electrolyte, which further impacted on overall impedance and capacitance in the system of symmetric supercapacitor. This manuscript also focused on the effect of gel polymer electrolyte thickness on the performance, with the optimum capacitance and rate capability obtained in GPE thickness of 12 μm. Comprehensive research contents which included the contrast on the variety of binder, electrolyte matrix, the thickness of GPE makes it superior for applying to practice in the research of symmetric supercapacitor. Nonetheless, there are some issues need to be clarified:
- There is a lack of supplementary materials, please upload.
- It is suggested to provide a brief background illustration aimed at conventional cellulose and Celgard separator in the part of matrices of electrolytes from the section of introduction.
- Thickness of Nitrocellulose filter membranes are approximately 170μm, which has a relatively large gap with the thickness interval of the GPE (from 5 μm to 36 μm). Is it scientific to compare performance directly?
- In the section of result discussion, there is only one subheading (Evaluation and optimization of GPE) which is difficult to summarize the whole section. Please subdivide or delete.
- The argument of choosing PVDF-HFP as binder instead of PVDF is not clear enough. Batteries with distinct binders, the matrix is also different in Fig.2f, the analysis “The specific capacitance of the GPE cells is slightly larger than the cell with PVDF binder” is not rational. Please modify or explain accordingly.
- Figure 1(c) has no relevant explanation in the text. What is the significance of its insertion? Please modify.
- Why the gel presented thicknesses of 2 μm and 5 μm after soaked in electrolyte porous? Did this phenomenon affect the performance?
- Please add annotations on the Fig.1f in order to distinguish nonaqueous electrolyte and electrodes.
- Please modify the SEM image format according to other papers in the journal.
- 2a-f involved thickness of gel, variety of electrolytes and binder, the discussion was not focused enough. Please modify.
- In the line 202-204, author speculated that the impedance of gel changed after activated is due to the improvement of ion-diffusion process, have the morphology and pore structure of the gel changed after activated?
- There is a brighter part in the upper right of Fig.3d, is it due to the aggregation of TEAPMo12 clusters.
- The overall organization of the article is not clear enough, the focus is not outstanding, please modify.

Author Response
- There is a lack of supplementary materials, please upload.
Thank you for pointing out the mistake. We failed to upload the file!! Sorry. We have included it at the end of the main manuscript now.
- It is suggested to provide a brief background illustration aimed at conventional cellulose and Celgard separator in the part of matrices of electrolytes from the section of introduction.
We have included the comparison between matrices in GPEs and conventional separators in the introduction part (highlighted) to stress the merits.
- Thickness of Nitrocellulose filter membranes are approximately 170μm, which has a relatively large gap with the thickness interval of the GPE (from 5 μm to 36 μm). Is it scientific to compare performance directly?
The referee is right in pointing this out. We didn’t have thinner nitrocellulose membranes. It is true that our cellulose membrane is thicker (and this is a detrimental factor) but it is also true that it has much larger pores (25 micro-meters pores) than our GPE (and in this case, this large pore size would be a favourable factor for cellulose). Thus, the referee is right that comparison is “shaky”. We have opted for warning the reader about these considerations when comparison between different separators are made (In discussion about Rct).
- In the section of result discussion, there is only one subheading (Evaluation and optimization of GPE) which is difficult to summarize the whole section. Please subdivide or delete.
Thank you for pointing it out. We add two more subheadings to organize the Results and Discussion part.
- The argument of choosing PVDF-HFP as binder instead of PVDF is not clear enough. Batteries with distinct binders, the matrix is also different in Fig.2f, the analysis “The specific capacitance of the GPE cells is slightly larger than the cell with PVDF binder” is not rational. Please modify or explain accordingly.
We believe the confusion comes from the fact that cells with Cellulose separator and with Cellgard separator included in Fig 2 f, BOTH have PVDF binder (whereas, our GPE cells do not have PVDF binder.
We apologize for the confusion and have clarified that point in the text.
Thus, we agree with the referee that “The specific capacitance of the GPE cells is slightly larger than the cell with PVDF binder.” is ambiguous. In Figure 2f, only 12 μm GPEs-based cells deliver slightly high capacitance. We have modified the sentence.
Concerning the argument of choosing PVDF-HFP instead of PVDF:
In Supplementary material 2 we present EIS of AC//PVDF-HFP binder electrode with cellulose separator. This cell shows an activation process too. Its Rct becomes smaller. The Rct (1.79 Ω) after activation is smaller than the AC//PVDF binder//cellulose separator cell (28.83 Ω).
Meanwhile, an AC// PVDF binder//PVDF-HFP GPE cell is characterized in Supplementary material 2. We observe the activation process as well. The Rct of this cell is smaller than that in conventional configuration (PVDF binder, Cellulose separator).
Thus, once PVDF-HFP exists in electrodes or separator (GPEs), the activation process exists, and Rct gets improved.
- Figure 1(c) has no relevant explanation in the text. What is the significance of its insertion? Please modify.
Our new text includes a short paragraph explaining the significance and intention in including this image (related to porosity). This explanation is included and highlighted in the revised manuscript.
- Why the gel presented thicknesses of 2 μm and 5 μm after soaked in electrolyte porous? Did this phenomenon affect the performance?
Actually all the PVDF-HFP films are porous (Fig. 1c). Those pores serve as the reservoir and the pathway for the electrolyte. However, we believe the referee refers to some inhomogeneities apparent in the photographs of Figure 1b. Those are bubbles, totally unrelated to the micropores in PVDF-HFP films. The bubbles accidentally sneaked into our 2 μm and 5 μm films because these films are too light to spread out smoothly. The gap under the films produced the bubbles after soaking. The bubbles can be removed when the films are squeezed in the electrolyte. It will not affect the performance of the devices because during the assembling the films are pressed with two electrodes in the electrolyte.
- Please add annotations on the Fig.1f in order to distinguish nonaqueous electrolyte and electrodes.
We have added two lines to separate the sandwich structure (Fig 1f): AC PVDF-HFP binder electrode/PVDF-HFP film(dry)/ AC PVDF-HFP binder electrode.
- Please modify the SEM image format according to other papers in the journal.
Absolutely right. We have modified the images. Thanks
- 2a-f involved thickness of gel, variety of electrolytes and binder, the discussion was not focused enough. Please modify.
Yes, there are many factors to correlate with properties and performance. We were consistent in analysing the same samples for all the measurements. Following the recommendation of the reviewer we have revised the text to further emphasize the main results, including also a new equivalent circuit.
- In the line 202-204, author speculated that the impedance of gel changed after activated is due to the improvement of ion-diffusion process, have the morphology and pore structure of the gel changed after activated?
This is an interesting question. It’s difficult to picture the morphology of soaked GPE. The pore structure of the gel should not change after activation. And we have included a sentence to indicate this assumption in our analysis. Instead, we found the activation process mainly involves the permeation of electrolyte in pores. We have applied a modified restricted finite diffusion element to fit the diffusion process, and updated the corresponding text.
- There is a brighter part in the upper right of Fig.3d, is it due to the aggregation of TEAPMo12 clusters.
This brighter zone is due to the surface topography of the large and irregular AC particle (in other words, brighter zones are at a higher level than darker ones). Further proof that POMs are not aggregated there is the fact that one can still observe bright white dots within the white background.
When POMs aggregate, they look like the image attached here
- The overall organization of the article is not clear enough, the focus is not outstanding, please modify.
We appreciate the reviewer comment. We think he/she is right because we have written a long section 3 (Results and discussion).
We have now re-structured that section into specific subsections properly labelled/titled.

Reviewer 2 Report
The work on the assembly of the compact supercapacitor seems to be interesting, however, the mentioned three supplementary materials in the main text of the manuscript is unavailable in the system.
Author Response
REVIEWER 2
The work on the assembly of the compact supercapacitor seems to be interesting, however, the mentioned three supplementary materials in the main text of the manuscript is unavailable in the system.
We apologize for this omission. Indeed, we failed to upload our supplementary data.
The error has been corrected. Thanks.

Reviewer 3 Report
The manuscript by Dr. Jun-Jie Zhu and co-workers describes a procedure to improve the overall capacitance by applying a new gel polymer electrolyte. The manuscript is well written and contains a very clear discussion and a detailed as well as exhaustive experimental protocol. The manuscript should be accepted for publication but before, some concerns should be revised:
- The abstract is too detailed and should be more concise: the authors should report only the most relevant outcomes and a brief introduction of their aims
- the subscript in the stoichiometric number 4 of “TEABF4” should be given
- as far as EIS characterization is concerned, authors claim that “All the spectra show a similar shape” providing the same circuit model to fit all experimental EIS data. This is not really true since it is clear that the 45°degree line which is expected from a “infinite-length” Warburg element is clearly confirmed only in the case of Cellgard. In the other cases, the ion diffusion approaches a gradual transition from the “infinite-length” Warburg to a “finite-length” Warburg for which, a vertical line is instead observed. It looks like that this feature is particularly relevant for the thinner films of 12 and 5 um. My suggestion is to compare the goodness of the fitting by including the “finite-length” Warburg element (also known as T element) instead of the “infinite-length” Warburg element already used in the study. This behavior may suggest that a more efficient ion diffusion inside the porous structure is observed for thinner films. In the other cases the ion diffusion is related to the bulk electrolyte.
Author Response
REVIEWER 3
The manuscript by Dr. Jun-Jie Zhu and co-workers describes a procedure to improve the overall capacitance by applying a new gel polymer electrolyte. The manuscript is well written and contains a very clear discussion and a detailed as well as exhaustive experimental protocol. The manuscript should be accepted for publication but before, some concerns should be revised:
We would like to thank the referee for his/her positive comments and insightful suggestions (below in comment 3), and no doubt they help us to improve the quality of the manuscript.
- The abstract is too detailed and should be more concise: the authors should report only the most relevant outcomes and a brief introduction of their aims
We have revised the abstract to make it shorter and more concise.
- the subscript in the stoichiometric number 4 of “TEABF4” should be given
All the “TEABF4” have been replaced by TEABF4
- as far as EIS characterization is concerned, authors claim that “All the spectra show a similar shape” providing the same circuit model to fit all experimental EIS data. This is not really true since it is clear that the 45°degree line which is expected from a “infinite-length” Warburg element is clearly confirmed only in the case of Celgard. In the other cases, the ion diffusion approaches a gradual transition from the “infinite-length” Warburg to a “finite-length” Warburg for which, a vertical line is instead observed. It looks like that this feature is particularly relevant for the thinner films of 12 and 5 um. My suggestion is to compare the goodness of the fitting by including the “finite-length” Warburg element (also known as T element) instead of the “infinite-length” Warburg element already used in the study. This behavior may suggest that a more efficient ion diffusion inside the porous structure is observed for thinner films. In the other cases the ion diffusion is related to the bulk electrolyte.
This is a very insightful suggestion. And we totally agree with the referee, because we followed his/her suggestion and that helped us to fit the EIS better with a simpler equivalent circuit and get a deeper understanding of our system. The modified restricted diffusion Ws has replaced semi-infinite Warburg diffusion in the equivalent circuit. There are three parameters in Ws. Ws-R is the diffusion impedance; Ws-P is an exponential factor: the number 0.5 means the infinite Warburg diffusion, and 1.0 means finite Warburg diffusion; Ws-T represents diffusion time. We have analysed the three parameters in detail, found that after the activation, diffusion impedance and diffusion time decrease. Meanwhile, Ws-P of 12 μm and 36 μm GPE cells increase remarkably, indicating the diffusion shifts from infinite diffusion in bulk electrolyte to finite diffusion in porous structure. Ws-P of 5 μm GPE cell is high, means it is dominated by finite diffusion initially, thanks to its thinnest thickness. And the diffusion in PVDF binder cell is infinite diffusion in the bulk electrolyte all the time. We have revised the corresponding discussion and highlighted it.
Thank you very much for your suggestion.

Round 2
Reviewer 2 Report
The paper presented the study on the coherent integration of the organic gel polymer electrolyte and polyoxometalate hybrid composite electrode to assemble compact supercapacitors. There are evident technological interest in the work, some comments are given below:
- What is the meaning of ‘their hydrophilic or hydrophilic nature’in line 47? or one of them should be ‘hydrophobic’?
- HPW12 and its hybrid electrode materials were also mentioned in the experimental part. However, no experimental result related with them was discussed in the paper. I am curious that, for both the polyoxometalate hybrid composites, which one is better for the high-performance supercapacitor?
- The optimized AC-GPE capacitor delivered a high volumetric capacitance and energy density. How much is the thickness of its electrode and the whole device, respectively?
- As shown in Fig 5d, the capacitor displayed excellent cyclic stability. How about the coulombic efficiency?
Author Response
The paper presented the study on the coherent integration of the organic gel polymer electrolyte and polyoxometalate hybrid composite electrode to assemble compact supercapacitors. There are evident technological interest in the work, some comments are given below:
We would like to thank the referee for his/her careful review. The answers to the comments are attached below:
- What is the meaning of ‘their hydrophilic or hydrophilic nature’in line 47? or one of them should be ‘hydrophobic’?
Thank you for pointing out the mistake. Yes, one of them should be hydrophobic. We have corrected it.
- HPW12 and its hybrid electrode materials were also mentioned in the experimental part. However, no experimental result related with them was discussed in the paper. I am curious that, for both the polyoxometalate hybrid composites, which one is better for the high-performance supercapacitor?
You are absolutely right. Sorry for this incongruity. Indeed, we initially prepared two hybrid materials (AC/TEAPW12 and AC/TEAPMo12). Considering gravimetric capacitance, AC/TEAPMo12 wins due to its relatively lower molecular weight (PMo12 1822 g/mol; PW12 2877 g/mol). But for volumetric capacitance, they show very little difference. Taking both (gravimetric and volumetric) features into account, and in order to focus on the main findings of our work, we decided to present only AC/TEAPMo12 as a case study for integrated electrode-GPE devices. We have deleted HPW12 and its hybrid electrode materials in experimental part to avoid misleading the reader.
- The optimized AC-GPE capacitor delivered a high volumetric capacitance and energy density. How much is the thickness of its electrode and the whole device, respectively?
The thickness of AC electrodes is around 65 μm (including 18 μm Al foil), and the thickness of AC/TEAPW12 electrodes is around 54 μm (including 18 μm Al foil). The loading masses of two kinds of electrodes are very close: around 2.4 mg/cm2. In the revised version, we have included the thickness and mass loading in experimental section. The thickness of the GPE electrolyte/separator is already indicated in the manuscript (12microm). We didn’t measure the thickness of the whole assembly but it should be very close to double that of the electrode plus the separator (i.e. ca. 120-140 micrometer).
On the other hand, concerning CR2032 coin cells cases to assemble full devices, the thickness of the devices is 3.2 mm. However, we didn’t use this thickness for calculation because the coin cells contain large dead space (spacers, springs). When calculating the power densities and energy densities, we take the whole volume of electrodes and GPEs (or separators) into account, because industrial coin cells are filled with multilayer electrodes and separators.
- As shown in Fig 5d, the capacitor displayed excellent cyclic stability. How about the coulombic efficiency?
For all the three capacitors, the coulombic efficiencies are higher than 99% after the first cycle. In the first cycle their coulombic efficiencies are relatively lower (97%) due to the consumption of some impurities.
We have added this information to our manuscript. Thanks

Reviewer 3 Report
Dear authors, thanks for your new revised version of the manuscript which is in line with my suggestions. please insert the equation that defines the Warburg element used to fit your EIS data: that can be done for instance in the caption of Table S2. It would be also usefull some references by the literature?
Author Response
Dear authors, thanks for your new revised version of the manuscript which is in line with my suggestions. please insert the equation that defines the Warburg element used to fit your EIS data: that can be done for instance in the caption of Table S2. It would be also useful some references by the literature?
Thank you for your kind suggestion. We totally agree that more details will help the readers who are not so acquainted with EIS. We have included the equation that defines the Warburg element in supplementary materials 6 where we put the other equations for calculation, and added relevant reference (ref 25-27) about the modified restricted diffusion.
